# Rethinking Learnable Tree Filter for Generic Feature Transform

**Lin Song**[1]    **Yanwei Li**[2]    **Zhengkai Jiang**[3]    **Zeming Li**[4]    **Xiangyu Zhang**[4]
**Hongbin Sun**[1*]    **Jian Sun**[4]    **Nanning Zheng**[1]

[1] College of Artificial Intelligence, Xi'an Jiaotong University
[2] The Chinese University of Hong Kong
[3] Institute of Automation, Chinese Academy of Sciences
[4] Megvii Inc. (Face++)
stevengrove@stu.xjtu.edu.cn, ywli@cse.cuhk.edu.hk, jiangzhengkai2017@ia.ac.cn,
{hsun, nnzheng}@mail.xjtu.edu.cn, {lizeming, zhangxiangyu, sunjian}@megvii.com

## Abstract

The Learnable Tree Filter presents a remarkable approach to model structure-preserving relations for semantic segmentation. Nevertheless, the intrinsic geometric constraint forces it to focus on the regions with close spatial distance, hindering the effective long-range interactions. To relax the geometric constraint, we give the analysis by reformulating it as a Markov Random Field and introduce a learnable unary term. Besides, we propose a learnable spanning tree algorithm to replace the original non-differentiable one, which further improves the flexibility and robustness. With the above improvements, our method can better capture long-range dependencies and preserve structural details with linear complexity, which is extended to several vision tasks for more generic feature transform. Extensive experiments on object detection/instance segmentation demonstrate the consistent improvements over the original version. For semantic segmentation, we achieve leading performance (82.1% mIoU) on the Cityscapes benchmark without bells-and-whistles. Code is available at https://github.com/StevenGrove/LearnableTreeFilterV2.

## 1   Introduction

In the last decade, the vision community has witnessed the extraordinary success of deep convolutional networks in various vision tasks [1–8]. However, in deep convolutional layers, the distribution of impact within an effective receptive field is found to be limited to a local region and converged to the gaussian [9], which brings difficulties to the long-range dependencies modeling. To address this problem, numerous local-based approaches [10–12] have been proposed to increase the receptive region of convolutional kernels by using pooling [13] or dilated operations [14]. Meanwhile, various global-based approaches [15–19] have been explored to aggregate features by modeling the pairwise relations based on the visual attention mechanism. However, there is still a conflict between long-range dependencies modeling and object details preserving.

Recently, the learnable tree filter module [20] (LTF-V1 module) tries to bridge this gap by performing feature aggregation on a minimum spanning tree. Since the minimum spanning tree is generated by the low-level guided features, it can retain the structural details with efficiency. Nevertheless, the *geometric constraint* and the *construction process* in the LTF-V1 module are found to be its Achilles' heels, which impede the usage for more generic feature transform. Firstly, the interactions with distant nodes along the spanning tree need to pass through nearby nodes, which brings the intrinsic *geometric constraint* to the tree filter. As illustrated in Fig. 1, this property forces the LTF-V1 module

---

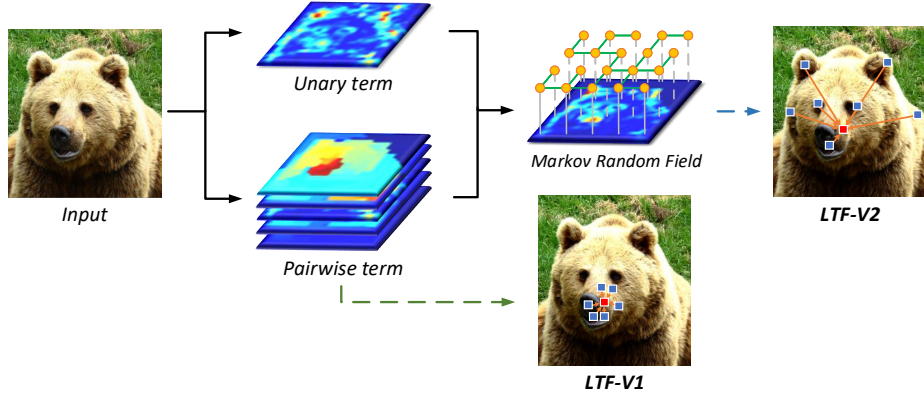

Figure 1: The comparison of the aggregation processes between the LTF-V1 module [20] and the proposed LTF-V2 module. One of the sink nodes is marked by the *red* square, and the corresponding source nodes with the *highest* filtering weights are presented as the *blue* squares. Limited by the intrinsic geometric constraint, the LTF-V1 can only give priority to the regions with close spatial distance. Different from it, our proposed LTF-V2 is more generic by adding a learnable unary term, which relaxes the geometric constraint and allows the filter to focus on the distant regions of interest.

to focus on the nearby regions. In addition, the *construction process* of the minimum spanning tree is non-differentiable, which is overly sensitive to the quality of guided features and prevents the LTF-V1 module from entirely learnable.

To remedy the shortages mentioned above, we rethink the learnable tree filter from the perspective of Markov Random Field [21] (MRF) and present the *Learnable Tree Filter V2 Module* (LTF-V2 module) for more generic feature transform. Specifically, it complements the LTF-V1 module by introducing a *learnable unary term* and a *learnable spanning tree algorithm*. The former one provides a modulation scalar [22–24] for each node, which can relax the geometric constraint and enable effective long-range interactions. Intuitively, as shown in the Fig. 1, the proposed unary term guides the LTF-V2 to focus on the coarsely distant regions of interest, and the tree-based pairwise term further refines the regions to fit original structures, bringing powerful semantic representations. Meanwhile, the proposed learnable spanning tree algorithm offers a simple and effective strategy to create a gradient tunnel between the spanning tree process and the feature transform module. This algorithm enables the LTF-V2 module to be more robust and flexible. Moreover, the LTF-V2 module maintains linear computational complexity and highly empirical efficiency of the LTF-V1 [2].

Overall, in this paper, we present a more generic method for structure-preserving feature transform with linear complexity. To demonstrate the effectiveness of the LTF-V2 module, we conduct extensive ablation studies on object detection, instance segmentation, and semantic segmentation. Both quantitative and qualitative experiments demonstrate the superiority of the LTF-V2 module over previous approaches. Even compared with other state-of-the-art works, the LTF-V2 module achieves competitive performance with much less resource consumption. Specifically, when applied to Mask R-CNN [25] (with ResNet50-FPN [26]), the LTF-V2 module obtains 2.4% and 1.8% absolute gains over the baseline on COCO benchmark for $AP^{box}$ and $AP^{seg}$ respectively, with negligible computational overhead. Meanwhile, the LTF-V2 module achieves **82.1**% mIoU on Cityscapes benchmark without bells-and-whistles, reaching leading performance on semantic segmentation.

## 2 Related Work

Since each neuron has limited effective receptive field, the deep convolutional networks [1, 3, 26] typically fail to capture long-range dependencies. Recently, to alleviate the problem, a large number of approaches have been proposed to model long-range context in existing deep convolutional models. These approaches can be generally divided into two categories, namely local-based and global-based.

The local-based approaches aim to aggregate the long-range context by increasing the local receptive region of each neuron, *e.g.*, pooling operation and dilated convolution. He *et al.* [27] and Zhao *et al.* [13] propose a spatial pyramid pooling for object detection and semantic segmentation, respectively. Yu *et al.* [28] introduces the dilated convolution to enlarge the receptive field of each convolution kernel explicitly. Dai *et al.* [12] further modifies the dilated convolution to a more generic form by replacing the grid kernels with the deformable ones. However, the modeling of pairwise relation by these methods is typically restricted to a local region and relies on the homogeneous prior.

The global-based approaches are mainly based on the attention mechanism, which is firstly applied in machine translation [29] as well as physical system modeling [30] and then extended to various vision tasks [15]. SENet [31], PSANet [32], and GENet [33] bring channel-wise relations to the network by performing down-sampling and attention in different channels. Non-Local [15] adopts the self-attention mechanism in the spatial domain to aggregate related features by generating the affinity matrix between each spatial node. It can model non-local relations but suffers from highly computational complexity. LatentGNN [18] and CCNet [16] are proposed to alleviate this problem by projecting features into latent space and stacking two criss-cross blocks, respectively.

Nevertheless, with the expansive receptive field, the detailed structure could not be preserved in the above methods. Our method bridges this gap by taking advantages of structure-aware spanning trees and global modeling of Markov Random Field.

## 3 Method

In this section, we theoretically analyze the deficiency of the LTF-V1 by reformulating it as a Markov Random Field and give a solution, namely the LTF-V2. Besides, we propose a learnable spanning tree algorithm to improve robustness and flexibility. With these improvements, we further present a new learnable tree filter module, called LTF-V2 module.

### 3.1 Notation and Problem Definition

Given an input feature map $X = \{x_i\}^N$ with $x_i \in \mathbb{R}^{1 \times C}$ and the corresponding guidance $G = \{g_i\}^N$ with $g_i \in \mathbb{R}^{1 \times C}$, where $N$ and $C$ indicate the number of input pixels and encoding channels, respectively. Specifically, the guidance $G$ gives the positions where the filter needs to preserve detailed structures. Following the original configuration [20, 34], we represent the topology of the guidance $G$ as a 4-connected planar graph, *i.e.*, $\mathcal{G} = \{\mathcal{V}, \mathcal{E}\}$, where $\mathcal{V} = \{\mathcal{V}_i\}^N$ is the set of nodes and $\mathcal{E} = \{\mathcal{E}_i\}^N$ is the set of edges. The weight of the edge reflects the feature distance between adjacent nodes. Our goal is to obtain the refined feature map $Y$ by transforming the input feature map $X$ with the property of guided graph $\mathcal{G}$, where the dimension of $Y$ is same with that of $X$.

### 3.2 Revisiting the Learnable Tree Filter V1

Recently, a learnable tree filter module [20] (LTF-V1 module) has been proposed as a flexible module to embed in deep neural networks. The procedure of the LTF-V1 module can be divided into two steps (the visualization is provided in the supplementary material).

First, a tree-based sparse graph $\mathcal{G}_T$ is sampled from the input 4-connected graph $\mathcal{G}$ by using the minimum spanning tree algorithm [35], as shown in Eq. 1. The pruning of the edges $\mathcal{E}$ gives priority to removing the edge with a large distance so that it can smooth out high-contrast and fine-scale noise while preserving major structures.

$$\mathcal{G}_T \sim \mathrm{MinimumSpanningTree}(\mathcal{G}). \qquad (1)$$

Second, we iterate over each node, taking it as the root of the spanning tree $\mathcal{G}_T$ and aggregating the input feature $X$ from other nodes. For instance, the output of the node $i$ can be formulated as Eq. 2, where $z_i$ is the normalizing coefficient and $E_{j,i}$ is the set of edges in the path from node $j$ to node $i$. Besides, $\omega_{k,m}$ indicates the edge weight between the adjacent nodes $k$ and $m$. $S_{\mathcal{G}_T}(\cdot)$ accumulate the edge weights along a path to obtain the *filtering weight*, *e.g.*, $\frac{1}{z_i} S_{\mathcal{G}_T}(\cdot)$ for the node $i$.

$$y_i = \frac{1}{z_i} \sum_{\forall j \in \mathcal{V}} S_{\mathcal{G}_T}(E_{j,i}) x_j, \quad \text{where } S_{\mathcal{G}_T}(E_{j,i}) = \exp(- \sum_{\forall (k,m) \in E_{j,i}} \omega_{k,m}). \qquad (2)$$

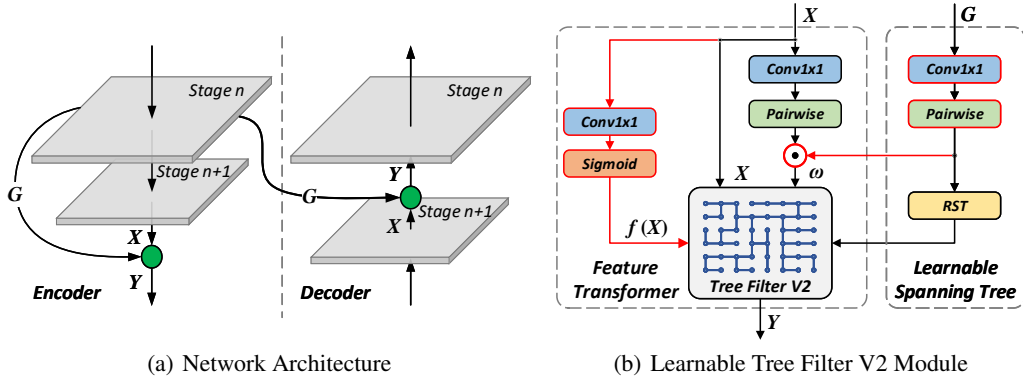

(a) Network Architecture       (b) Learnable Tree Filter V2 Module

Figure 2: The diagram of the network architecture and the proposed Learnable Tree Filter V2 module. The *green* circles in (a) denote the LTF-V2 modules, which generate the spanning tree with the low-level feature (*i.e.*, "Stage n") and then transform the high-level feature (*i.e.*, "Stage n+1"). We use *red* in (b) to represent the newly added components compared with the LTF-V1 module. $f(X)$ and $\omega$ are the elements in unary and pairwise terms, respectively. "RST" denotes the proposed close random spanning tree algorithm. And "Pairwise" indicates calculating the distance for each edge.

In the original design [20], the edge weight is instantiated as the euclidean distance between features of adjacent nodes, *e.g.*, $\omega_{k,m} = |x_k - x_m|^2$. In addition, due to the tree structure is acyclic, we can use dynamic programming algorithms to improve computational efficiency significantly.

## 3.3 Learnable Tree Filter V2

Extensive experiments [20] demonstrate the effectiveness of LTF-V1 module on semantic segmentation. However, the performance of the LTF-V1 module in the instance-aware tasks is unsatisfactory, which is inferior to some related works (*e.g.*, LatentGNN [18] and GCNet [19]). The details are presented in Tab. 1. We speculate that the reason can be attributed to the intrinsic geometric constraint. In this section, we try to analyze this problem and give the solution.

**Modeling.** According to the Eq. 2, we can consider the LTF-V1 as the statistical expectation of the sampling from input features under a specific distribution. The distribution is calculated by using the spanning tree. To be more generic, we first define a set of random latent variables as $\mathbf{H} = \{h_i\}^N$ with $h_i \in \mathcal{V}$. And we give a more generic form of feature transform, which is elaborated in Eq. 3. For instance, when the probability distribution $P_{\mathcal{G}_T}$ is set to the filtering weight of the LTF-V1, it will be equivalent to the LTF-V1.

$$y_i = \mathbb{E}_{h_i \sim P_{\mathcal{G}_T}}[x_{h_i}] = \sum_{\forall j \in \mathcal{V}} P_{\mathcal{G}_T}(h_i = j)x_j. \tag{3}$$

Moreover, we adopt the Markov Random Field (MRF) [21] to model the distribution $P_{\mathcal{G}_T}$, which is a powerful tool to model the joint distribution of latent variables by defining both unary and pairwise terms. The formulation of the MRF is shown in Eq. 4, where $\mathbf{O}$ is the set of random observable variables associating with the input feature. $\phi_i$ and $\psi_{i,j}$ represent unary and pairwise terms of the MRF, respectively. Besides, $Z$ indicates the partition function.

$$P_{\mathcal{G}_T}(\mathbf{H}|\mathbf{O} = X) = \frac{1}{Z} \prod_{\forall i \in \mathcal{V}} \phi_i(h_i, x_i) \prod_{\forall (i,j) \in \mathcal{E}} \psi_{i,j}(h_i, h_j). \tag{4}$$

**Analysis.** First of all, we analyze why the LTF-V1 module works well on semantic segmentation. By comparing the formulas, we give a specific MRF on tree $\mathcal{G}_T$, which is formally equivalent to the LTF-V1 (the proof is provided in the supplementary material). Specifically, the formulation is shown as Eq. 5 and Eq. 6, where $\delta(\cdot)$ denotes the unit impulse function and $\mathrm{Desc}_{\mathcal{G}_T}(i, j)$ is the descendants of node $j$ when node $i$ is the root of tree $\mathcal{G}_T$.

$$\phi_i(h_i, x_i) \equiv 1, \tag{5}$$

$$\psi_{i,j}(h_i, h_j) := \begin{cases} \delta(h_i - h_j) & h_i \notin \text{Desc}_{\mathcal{G}_T}(i,j) \\ \exp(-\omega_{i,j})\delta(h_i - h_j) & h_i \in \text{Desc}_{\mathcal{G}_T}(i,j) \end{cases} \tag{6}$$

The MRF is demonstrated to preserve the local structure by modeling data-dependent pairwise terms between adjacent nodes in previous works [21, 36–39]. This property of MRF is extremely desired by semantic segmentation [40–43]. Furthermore, combined with the visualizations with rich details [20], we consider the reason for the effectiveness of LTF-V1 on semantic segmentation lies in modeling learnable pairwise terms.

We further analyze why the LTF-V1 module performs unsatisfactorily in the instance-aware tasks. The effective long-range dependencies modeling, which provides powerful semantic representations, is proved to be crucial for instance-aware tasks [15–18]. However, due to the intrinsic geometric constraint of the LTF-V1, as illustrated in Fig. 1, interactions with distant nodes need to pass through nearby nodes. Meanwhile, as shown in Eq. 5, the LTF-V1 models the unary term as a constant scalar, resulting in the filtering weight monotonously decreases as the distance increase along a path (the proof is provided in the supplementary material). Therefore, these properties force the LTF-V1 to focus on the **nearby region**, leading to the difficulty of long-range interactions and unsatisfactory performance in instance-aware tasks.

**Solution.** Eventually, we try to give a solution to address the problem. To focus on the distant region, we need to relax the geometric constraint and allow the filtering weight of distant nodes to be larger than that of close ones. In this paper, motivated by confidence estimation methods [22–24], we present a learnable unary term which can meet the requirements (the proof is provided in the supplementary material). The formulation is shown on Eq. 7, where $f(\cdot)$ denotes a unary function embedded in the deep neural network and $\beta$ is a learnable parameter. Intuitively, for a node, $f(\cdot)$ reflects the confidence of its input feature, while $\beta$ is the potential for choosing other features.

$$\phi_i(h_i, x_i) := \begin{cases} f(x_i) & h_i = i \\ \exp(-\beta) & h_i \neq i \end{cases} \tag{7}$$

Let Eq. 7 and Eq. 6 substitute into Eq. 4. Since the tree $\mathcal{G}_T$ is a acyclic graph, the closed-form solution of Eq. 3 can be derived as Eq. 8 by using the belief propagation algorithm [44]. $z_i$ is the normalizing coefficient and $|E_{j,i}|$ denotes the number of edges in the path.

$$y_i = \frac{1}{z_i} \sum_{\forall x_j \in X} S_{\mathcal{G}_T}(E_{j,i}) \exp(-\beta)^{|E_{j,i}|} f(x_j) x_j. \tag{8}$$

Accordingly, we define the Eq. 8 as a more generic form of the learnable tree filter, namely the **Learnable Tree Filter V2 (LTF-V2)**. The unary and pairwise terms in the LTF-V2 are complementary and promoting to each other. The unary term on Eq. 7 allows the LTF-V2 to focus on the **distant region**, bringing more effective long-range interactions. Meanwhile, the data-dependent pairwise term can further refine the distant region to fit detailed structures.

### 3.4 Learnable Random Spanning Tree

Although the LTF-V1 module takes the first step to make the tree filtering process [34] trainable, the inside minimum spanning tree algorithm is still not differentiable. This problem prevents it from being entirely learnable. Moreover, the topology of the spanning tree is determined by guided features only. Therefore, the initialization and source selection of guided features could significantly impact the performance in the original design. In this paper, we bridge this gap by proposing a simple architecture and a *close random-spanning-tree* algorithm, which is briefly introduced in Fig. 2.

Firstly, we propose a simple strategy to make the spanning tree process learnable. As shown in the right dashed rectangle of Fig. 2(b), we calculate the joint affinities by performing the element-wise production on pairwise similarities, which are generated from the input feature $X$ and the guided feature $G$, respectively. And then, we use the joint affinities as the edge weights $\omega$ for the feature transformer. This strategy creates a gradient tunnel between the guided feature and the output feature $Y$ to make the guided feature trainable utilizing the back-propagation algorithm [45].

Moreover, a close random spanning tree algorithm is designed to replace the original minimum spanning tree in the training phase. As illustrated in Alg. 1, the proposed algorithm is a modification

**Algorithm 1:** Close random spanning tree

---

**Input:** A 4-connected graph $\mathcal{G}$.
**Output:** Random spanning tree $\mathcal{G}_T$.

1   $\mathcal{G}_T \leftarrow \emptyset$.
2   **for** $e \in E(\mathcal{G})$ **do**
3      $l(e) \leftarrow e$.                          ▷ Initialize a label for each edge.
4   **while** $|V(\mathcal{G})| > 1$ **do**
5      **for** $v_i \in V(\mathcal{G})$ **do**
6          $e_i \sim E_{\mathcal{G}}(v_i)$.                   ▷ Sample an incident edge.
7          $\mathcal{G}_T \leftarrow \mathcal{G}_T \cup \{l(e_i)\}$.
8      $\mathrm{Contract}(E(\mathcal{G}))$.                ▷ Contraction algorithm.
9      $\mathrm{Flatten}(\mathcal{G})$.           ▷ Remove loops and parallel edges.
10 **return** $\mathcal{G}_T$.

---

of the Boruvka's algorithm [46], which replaces the minimum selection to the stochastic sampling (line 6 in Alg. 1) according to the edge weights. This algorithm has the ability to regularize the network and avoid falling into local optima, which is resulted from the greedy strategy of the minimum spanning tree. For the reason that the input $\mathcal{G}$ is a 4-connected planar graph, the computational complexity of the proposed algorithm can be reduced to linear on the number of pixels by using the edge contraction operation [47] (line 8 in Alg. 1). In the training phase, the weights of selected edges as well as the sampling distribution will be optimized. Besides, in the inference phase, we still use the minimum spanning tree to keep the results deterministic.

### 3.5   Network Architecture

Based on the algorithms in Sec. 3.3 and Sec. 3.4, we propose a generic feature transform module, namely the Learnable Tree Filter V2 module (*LTF-V2 module*). As illustrated in Fig. 2(b), the LTF-V2 module is composed of a feature transformer and a learnable spanning tree process. To highlight the effectiveness of the LTF-V2 module, we adopt a simple embedding operator to instantiate the function $f(\cdot)$ in the unary term, *i.e.*, $f(x_i) = \mathrm{Sigmoid}(\pi x_i^\top)$, where $\pi \in \mathbb{R}^{1 \times C}$. In addition, following the design of the LTF-V1 module, we instantiate the pairwise function $S_{\mathcal{G}_T}(\cdot)$ of edge weights as the *Euclidean distance* and adopt the *grouping strategy* for the edge weights and the unary term. Specifically, the grouping strategy [48] is adopted to split the input feature into specific groups along the channel dimension and aggregate them with different weights. The default group number is set to 16, and the detailed comparison is provided in the supplementary material.

Due to the linear computational complexity, the LTF-V2 module is highly efficient and can be easily embedded as a learnable layer into deep neural networks. To this end, we propose two usages of the module for the encoder and the decoder, respectively. The usage for the encoder is shown as the left part of Fig. 2(a), which embeds the LTF-V2 module between adjacent stages of the encoder. In this way, the resize operation is used on the low-level feature (*e.g.*, the "Stage n" in Fig. 2(a)) when the dimensions of the low-level feature and the high-level feature (*e.g.*, the "Stage n+1" in Fig. 2(a)) are inconsistent. Furthermore, the LTF-V2 module adopts the resized low-level feature as a guided feature $G$ to generate the spanning tree and transform the high-level feature. Different from the encoder, the usage for the decoder is shown in the right diagram of Fig. 2(a). The LTF-V2 module is embedded in the decoder and relies on the corresponding low-level feature in the encoder to generate the spanning tree.

In this paper, we conduct experiments on semantic segmentation and instance-aware tasks, *i.e.*, object detection and instance segmentation. For semantic segmentation, the ResNet [26] is adopted as the backbone with the naive decoder following the LTF-V1. While for instance-aware tasks, we adopt the Mask R-CNN [25] with FPN as the decoder and ResNet [26]/ResNeXt [48] as the encoder.

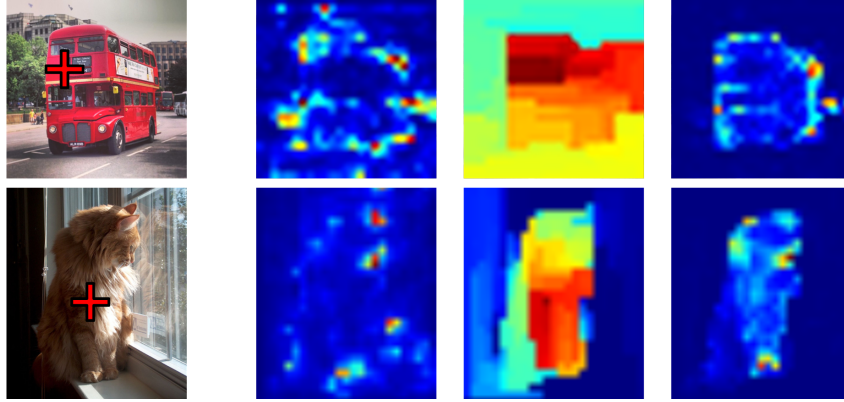

Figure 3: Visualization of the affinity maps in specific positions (marked by the *red* cross in each image). The affinity maps (from left to right) are generated from the *unary term*, the *pairwise term* and the *LTF-V2*, respectively.

## 4 Experiments

To demonstrate the effectiveness of the proposed method, several experiments are conducted on *COCO* [49] for object detection/instance segmentation (instance-aware tasks) and *Cityscapes* [50] for semantic segmentation.

### 4.1 Experiments in Instance-aware Tasks

#### 4.1.1 Training Setting

Following the configuration of Mask R-CNN in the Detectron2 [51] framework, we employ the FPN based decoder and a pair of 4-convolution heads for bounding box regression and mask prediction, respectively. All the backbones are pre-trained on the ImageNet classification dataset [52] unless otherwise specified. In the training phase, input images are resized so that the shorter side is 800 pixels. All the hyper-parameters are identical to the 1x schedule in the Detectron2 framework.

#### 4.1.2 Ablation Study

**Components of the LTF-V2.** Intuitively, as illustrated in Fig. 3, the properties of unary and pairwise terms are complementary. The unary term can guide the LTF-V2 to focus on the coarsely distant regions, and the pairwise term further refines the regions to fit the detailed structures. Furthermore, we give the qualitative comparisons on Tab. 4, which demonstrates the balance of the two terms can work better than the single one. To reveal the impact of the data-dependent spanning tree, we adopt the uniformly random spanning tree [47] for the pairwise term by default.

**Stages.** We explore the effect when inserting the LTF-V2 module after the last layer of different stages. The performance in stage 4 is better than in stage 3 and stage 5, which reflects the advantage of using both semantic context and detailed structure. Moreover, due to the efficiency of our proposed framework, we can further insert the LTF-V2 module into multiple stages to fuse multi-scale features. In this way, as shown in Tab. 1, our method achieves 41.2% on $\mathrm{AP^{box}}$ and 37.0% on $\mathrm{AP^{seg}}$, which has **2.4**% and **1.8**% absolute gains over the baseline on $\mathrm{AP^{box}}$ and $\mathrm{AP^{seg}}$, respectively.

**Spanning tree algorithm.** To evaluate the proposed learnable spanning tree algorithm, we conduct ablation studies with the original minimum spanning tree algorithm. As shown in Tab. 2, the results demonstrate the effectiveness of the proposed learnable spanning tree algorithm, which achieves consistent improvements on both localization and segmentation. Especially *without* pre-training, the improvement is more prominent, which indicates that the proposed algorithm improves the robustness to alleviate the adverse impact of random initialization.

**Stronger backbones.** To further validate the effectiveness, we evaluate the proposed LTF-V2 modules on stronger backbones. As shown in Tab. 3, we adopt ResNet-101 or ResNeXt-101 as the

Table 1: Comparisons among related works and the LTF-V2 module on different stages for COCO 2017 *val* set. All experiments are conducted on the Mask-RCNN framework with 1x learning rate schedule. GCNet [19] is used after each bottleneck, and other blocks are applied at the end of specific stages in the backbone. The settings of all the models are configured as the original papers.

| Model | Stage | $AP_{box}$ | $AP_{box}^{50}$ | $AP_{box}^{75}$ | $AP_{seg}$ | $AP_{seg}^{50}$ | $AP_{seg}^{75}$ | #FLOPs | #Params |
|---|---|---|---|---|---|---|---|---|---|
| ResNet-50 (1x) | - | 38.8 | 58.7 | 42.4 | 35.2 | 55.6 | 37.6 | 279.4B | 44.4M |
| +Non-Local [15] | 4 | 39.5 | 59.6 | 42.7 | 35.6 | 56.7 | 37.6 | +10.67B | +2.09M |
| +CCNet [16] | 345 | 40.1 | 60.4 | 44.1 | 36.0 | 57.4 | 38.4 | +16.62B | +6.88M |
| +LatentGNN [18] | 345 | 40.6 | 61.3 | 44.5 | 36.6 | 58.1 | 39.2 | +3.59B | +1.07M |
| +GCNet [19] | All | 40.7 | 61.0 | 44.2 | 36.7 | 58.1 | 39.2 | +0.35B | +10.0M |
| +LTF-V1 | 345 | 40.0 | 60.4 | 43.7 | 36.1 | 57.5 | 38.4 | +0.31B | +0.06M |
| +LTF-V2 | 3 | 40.1 | 59.9 | 43.9 | 36.0 | 57.1 | 38.3 | +0.43B | +0.02M |
| | 4 | 40.6 | 61.0 | 44.4 | 36.6 | 58.2 | 39.0 | +0.26B | +0.04M |
| | 5 | 40.2 | 60.5 | 43.6 | 36.1 | 57.5 | 38.4 | +0.17B | +0.08M |
| | 345 | **41.2** | **61.6** | **45.2** | **37.0** | **58.4** | **39.5** | +0.68B | +0.14M |

Table 2: Comparison of different spanning tree algorithms for the LTF-V2 module on COCO 2017 *val* set. **MST** and **LST** are the minimum spanning tree algorithm and the proposed learnable spanning tree algorithm, respectively. **Pretrain** indicates the pre-trained weights for the backbone.

| Pretrain | MST | LST | $AP_{box}$ | $AP_{seg}$ |
|---|---|---|---|---|
| Scratch | ✓ | ✗ | 29.4 | 26.7 |
| | ✗ | ✓ | **30.3** | **27.6** |
| ImageNet [52] | ✓ | ✗ | 40.9 | 36.8 |
| | ✗ | ✓ | **41.2** | **37.0** |

Table 3: Comparisons among different backbones for the Mask-RCNN framework on COCO 2017 *val* set.

| Model | $AP_{box}$ | $AP_{seg}$ |
|---|---|---|
| ResNet-101 (1x) | 40.7 | 36.6 |
| +LTF-V1 | 41.6 | 37.3 |
| +LTF-V2 | **42.5** | **38.0** |
| ResNeXt-101 (1x) | 43.0 | 38.3 |
| +LTF-V1 | 43.8 | 39.0 |
| +LTF-V2 | **44.5** | **39.7** |
| ResNeXt-101 + Cascade (1x) | 45.5 | 39.3 |
| +LTF-V1 | 46.2 | 40.0 |
| +LTF-V2 | **46.9** | **40.4** |

backbone. Following the same strategy, we insert the learnable tree filter modules at the end of stage3, stage4, and stage5, respectively. The LTF-V2 module still achieves noticeable performance gains over stronger baselines. Specifically, when using cascade strategy [53] and ResNeXt-101 as the backbone, we achieve 1.4% and 1.1% absolute gains over the baseline for $AP^{box}$ and $AP^{seg}$, respectively.

## 4.2 Experiments on Semantic Segmentation

### 4.2.1 Training Setting

For semantic segmentation, training images are randomly resized by 0.5 to 2.0× and cropped to 1024×1024. The random flipping horizontally is applied for data augmentation. Furthermore, we employ 8 GPUs for training, and the effective mini-batch size is 16. Following conventional protocols [20, 31, 43], we set the initial learning rate to 0.01 and employ the "poly" schedule (*i.e.*, multiply $(1 - \frac{iter}{max\_iter})^{power}$ for each iteration) with $power = 0.9$. All models are optimized by using synchronized SGD with the weight decay of 0.0001 and the momentum of 0.9. For fair comparisons, we adopt the LTF-V2 modules in each stage of the decoder as the LTF-V1 does.

### 4.2.2 Ablation Study

As shown in Tab. 5, we give quantitative comparisons between the LTF-V1 module and the LTF-V2 module. The results show that both yield significant improvement over the baseline. Specifically, without data augmentation, the LTF-V2 module achieves **4.5**% and **1.5**% absolute gains on mIoU over the baseline and the LTF-V1 module, respectively. Moreover, we adopt multi-scale and flipping augmentations for testing. The performance of the LTF-V2 module is further improved, which still attains **3.4**% and **1.7**% absolute gains on mIoU over the baseline and the LTF-V1 module, respectively.

Table 4: Comparisons among different components of the LTF-V2 module on COCO 2017 *val* set. **Unary** and **Pairwise** are the components of the proposed MRF, where the pairwise term adopts the uniformly random spanning tree by default. **LST** is the proposed learnable spanning tree algorithm.

| Model | Unary | Pairwise | LST | $AP_{box}$ | $AP_{box}^{50}$ | $AP_{box}^{75}$ | $AP_{seg}$ | $AP_{seg}^{50}$ | $AP_{seg}^{75}$ |
|---|---|---|---|---|---|---|---|---|---|
| | ✗ | ✗ | ✗ | 38.8 | 58.7 | 42.4 | 35.2 | 55.6 | 37.6 |
| | ✓ | ✗ | ✗ | 39.7 | 59.7 | 43.5 | 35.8 | 56.8 | 38.1 |
| | ✗ | ✓ | ✗ | 39.6 | 59.5 | 43.3 | 35.7 | 56.6 | 37.9 |
| ResNet-50 (1x) | ✓ | ✓ | ✗ | 40.7 | 61.2 | 44.8 | 36.7 | 58.2 | 39.3 |
| | ✗ | ✓ | ✓ | 40.2 | 60.7 | 43.8 | 36.2 | 57.4 | 38.8 |
| | ✓ | ✓ | ✓ | **41.2** | **61.6** | **45.2** | **37.0** | **58.4** | **39.5** |

Table 5: The ablation studies conducted on Cityscapes *val* set. **MS** and **Flip** denote adopting multi-scale and flipping augmentation for testing, respectively.

| Model | MS&Flip | mIoU (%) | mAcc (%) |
|---|---|---|---|
| ResNet-50 | ✗ | 72.9 | 95.5 |
| +LTF-V1 | ✗ | 75.9 | 95.8 |
| +LTF-V2 | ✗ | **77.4** | **96.0** |
| ResNet-50 | ✓ | 75.5 | 95.9 |
| +LTF-V1 | ✓ | 77.2 | 96.0 |
| +LTF-V2 | ✓ | **78.9** | **96.1** |

Table 6: Comparisons with state-of-the-art results on Cityscapes *test* set. Our model is trained with fine annotations only.

| Model | Backbone | mIoU (%) |
|---|---|---|
| PSPNet [13] | ResNet-101 | 78.4 |
| DFN [43] | ResNet-101 | 79.3 |
| DenseASPP [54] | DenseNet-161 | 80.6 |
| LTF-V1 [20] | ResNet-101 | 80.8 |
| CCNet [16] | ResNet-101 | 81.4 |
| DANet [55] | ResNet-101 | 81.5 |
| SPNet [56] | ResNet-101 | 82.0 |
| Ours (LTF-V2) | ResNet-101 | **82.1** |

### 4.2.3 Comparison with State-of-the-arts

To further improve the performance, we adopt a global average pooling operation and an additional ResBlocks [26] in the decoder, which follows the design in the LTF-V1 module. Specifically, an extra global average pooling operator is inserted at the end of stage4 in the encoder. In addition, multiple ResBlocks based on "Conv3×3" are added before each upsampling operator in the decoder. Similar with the conventional protocols [20, 31, 43], we further finetune our model on both *train* and *val* sets. As shown in Tab. 6, we achieve **82.1%** mIoU result on Cityscapes *test* set, which is superior to other state-of-the-art approaches with ResNet-101 backbone and only fine annotations.

## 5 Conclusion

In this paper, we first rethink the advantages and shortages of the LTF-V1 by reformulating it as a Markov Random Field. Then we present a learnable unary term to relax the geometric constraint and enable effectively long-range interactions. Besides, we propose the learnable spanning tree algorithm to replace the non-differentiable one for an entirely learnable tree filter. Extensive ablation studies are conducted to elaborate on the effectiveness and efficiency of the proposed method, which is demonstrated to bring significant improvements on both instance-aware tasks and semantic segmentation with negligible computational overhead. We hope that the perspective of Markov Random Field for context modeling can provide insights into future works, and beyond.

## Broader Impact

Context modeling is a powerful tool to improve the ability for feature representation, which has been widely applied in real-world scenarios, *e.g.*, computer vision and natural language processing. The traditional tree filter [34] already has great impacts on many low-level computer vision tasks, owing to its structure-preserving property and high efficiency. This paper further releases its representation potential by relaxing the geometric constraint. Specifically, our method provides a new perspective for context modeling by unifying the learnable tree filter with the Markov Random Field, which is further demonstrated to be effective in several vision tasks with negligible computational and

parametric overheads. These properties of our method have great potentials, which allow our method and principle to extend to other complex tasks with large-number nodes, *e.g.*, replacing the attention module of transformer for natural language processing and enhancing sequential representation for video analysis.

## Acknowledgments and Disclosure of Funding

This research was supported by National Key R&D Program of China (No. 2017YFA0700800), National Natural Science Foundation of China (No. 61790563 and 61751401) and Beijing Academy of Artificial Intelligence (BAAI).

## Footnotes

[2]The runtime benchmark is provided in the supplementary material.

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
