[Supplementary Material]

# Supplementary Material

## A   Runtime

Due to trees are acyclic graphs, we can adopt a well-designed dynamic programming algorithm to reduce the computational complexity to linear w.r.t vertex number. Besides, in order to improve the efficiency on GPU devices, we parallelize the algorithm along with batches, channels, and nodes of the same depth. As shown in Fig. 4, the empirical runtime of our method is far less than that of the Non-Local network [12], which uses an extremely sophisticated tensor library. We believe that the efficiency of LTF-V2 can be further improved through device-oriented code optimization.

Figure 4: Comparisons of the runtime on a Tesla V100 GPU among LTF-V1, LTF-V2 and Non-Local network. The number of feature channels is set to 512

## B   Visualization

In this section, as shown in Fig. 5 and Fig. 6, we present several visualization results of the baseline (w/o context block) and the proposed LTF-V2 module for instance-aware tasks and semantic segmentation, respectively.

**Groudtruth**                **Baseline**                **LTF-V2**

Figure 5: Visualization results on COCO 2017 *val* set. We use different colors to distinguish between instances that do *not* represent categories. The results demonstrate the superiority of the LTF-V2 module in terms of both semantics and details

**Image**        **Baseline**        **LTF-V2**

Figure 6: Visualization results on Cityscapes *val* set. The red dotted ellipses indicate the area of the major difference between the baseline and the LTF-V2 module

# C   Additional Ablation Study

Table 7: Comparisons among different positions and different groups on COCO 2017 *val* set when using ResNet-50 as our backbone. **Enc** and **Dec** denote embedding the blocks in the backbone and the FPN, respectively

| Model | Position | Groups | $AP_{box}$ | $AP_{box}^{50}$ | $AP_{box}^{75}$ | $AP_{seg}$ | $AP_{seg}^{50}$ | $AP_{seg}^{75}$ | #FLOPs | #Params |
|---|---|---|---|---|---|---|---|---|---|---|
| +LTF-V1 | Dec | 16 | 39.6 | 60.0 | 43.4 | 35.8 | 56.9 | 38.0 | +0.45B | +0.02M |
|  | Enc | 1 | 39.6 | 59.6 | 43.1 | 35.6 | 56.5 | 37.9 | +0.09B | +0.01M |
|  | Enc | 16 | 40.0 | 60.4 | 43.7 | 36.1 | 57.5 | 38.4 | +0.31B | +0.06M |
| +LTF-V2 | Dec | 16 | 39.9 | 60.1 | 43.6 | 36.0 | 57.1 | 38.4 | +1.23B | +0.06M |
|  | Enc | 1 | 40.2 | 60.5 | 44.0 | 36.3 | 57.5 | 38.8 | +0.13B | +0.01M |
|  | Enc | 4 | 40.9 | 61.3 | 44.5 | 36.8 | 58.4 | 39.4 | +0.24B | +0.04M |
|  | Enc | 8 | 40.9 | 61.4 | 44.3 | 36.8 | 58.4 | 39.2 | +0.39B | +0.07M |
|  | Enc | 16 | **41.2** | **61.6** | **45.2** | **37.0** | 58.4 | **39.5** | +0.68B | +0.14M |
|  | Enc | 32 | 41.1 | 61.6 | 44.8 | 37.0 | **58.7** | 39.4 | +1.28B | +0.29M |

(a)  Affinity maps of LTF-V1 (top) and LTF-V2 (bottom) with different groups

(b)  Affinity maps of LTF-V2 for different instances

(c)  Affinity maps of LTF-V2 in different stages

Figure 7: Visualization of the affinity maps in specific positions (marked by the red cross in each image). The responses of different groups, instances, and stages are illustrated in (a), (b), and (c), respectively. The figure (a) intuitively shows that the diversity of the LTF-V2 module is significantly higher than that of the LTF-V1 module due to the relaxation of geometric constraints. The figure (b) reveals that the LTF-V2 module is aware of the structure of instance, even when the instances of the same category are overlapping. Besides, the heat maps (from left to right) in (c) correspond to stage3, stage4, and stage5 in the encoder, respectively.

# D  Algorithm Proof

In this section, we present the detailed proofs for the claims about the learnable tree filter. Please note that the symbols follow the definitions in the main paper.

**Lemma 1.** *Given a Markov Random Field on the tree $\mathcal{G}_T$, whose unary and pairwise terms are the Eq. 9 and the Eq. 10 respectively, the marginal probability of latent variable satisfies $P_{\mathcal{G}_T}(h_i = j) = \frac{1}{z_i} S_{\mathcal{G}_T}(\mathbf{E}_{j,i})$.*

$$\phi_i(h_i, x_i) \equiv 1, \tag{9}$$

$$\psi_{i,j}(h_i, h_j) := \begin{cases} \delta(h_i - h_j) & h_i \notin \text{Desc}_{\mathcal{G}_T}(i,j) \\ \exp(-\omega_{i,j})\delta(h_i - h_j) & h_i \in \text{Desc}_{\mathcal{G}_T}(i,j) \end{cases} \tag{10}$$

*Proof.* To obtain the marginal probability of the Markov Random Field on an acyclic graph, we adopt the belief propagation algorithm as shown on Eq. 11, where $\mathcal{N}_i$ denotes the set of adjacent nodes of node $i$ in the tree.

$$P_{\mathcal{G}_T}(h_i) = \frac{1}{z_i}\phi_i(h_i, x_i) \prod_{\forall j \in \mathcal{N}_i} m_{j,i}(h_i),$$
$$m_{j,i}(h_i) = \sum_{\forall h_j \in \mathcal{V}} \phi_j(h_j, x_j)\psi_{j,i}(h_j, h_i) \prod_{\forall k \in \mathcal{N}_j \setminus i} m_{k,j}(h_j). \tag{11}$$

For $h_i = i$, the marginal probability is

$$\begin{aligned}
P_{\mathcal{G}_T}(h_i = i) &= \frac{1}{z_i} \prod_{\forall j \in \mathcal{N}_i} m_{j,i}(i) \\
&= \frac{1}{z_i} \prod_{\forall j \in \mathcal{N}_i} \left( \sum_{\forall h_j \in \mathcal{V}} \delta(h_j - i) \prod_{\forall k \in \mathcal{N}_j \setminus i} m_{k,j}(h_j) \right) \\
&= \frac{1}{z_i} \prod_{\forall j \in \mathcal{N}_i} \prod_{\forall k \in \mathcal{N}_j \setminus i} m_{k,j}(i) \\
&= \frac{1}{z_i} \prod_{\forall j \in \mathcal{N}_i} \prod_{\forall k \in \mathcal{N}_j \setminus i} \prod_{\forall u \in \mathcal{N}_k \setminus k} \cdots 1 \\
&= \frac{1}{z_i}.
\end{aligned} \tag{12}$$

For $h_i = u$ and $u \neq i$, the marginal probability is

$$\begin{aligned}
P_{\mathcal{G}_T}(h_i = u) &= \frac{1}{z_i} \prod_{\forall j \in \mathcal{N}_i \cap \{p | x_u \in \text{Desc}(i,p)\}} m_{j,i}(u) \\
&= \frac{1}{z_i} \sum_{\forall h_j \in \mathcal{V}} \psi_{j,i}(h_j, u) \prod_{\forall k \in \mathcal{N}_j \setminus i} m_{k,j}(h_j) \\
&= \frac{1}{z_i} \sum_{\forall h_j \in \mathcal{V}} \exp(-\omega_{j,i})\delta(h_j - u) \prod_{\forall k \in \mathcal{N}_j \setminus i} m_{k,j}(h_j) \\
&= \frac{1}{z_i}\exp(-\omega_{j,i}) \prod_{\forall k \in \mathcal{N}_j \setminus i} m_{k,j}(u) \\
&= \frac{1}{z_i}\exp(-\omega_{j,i})\exp(-\omega_{k,j}) \cdots \prod_{\forall v \in \mathcal{N}_u \setminus par(u)} m_{v,u}(u) \\
&= \frac{1}{z_i}\exp(-\omega_{j,i})\exp(-\omega_{k,j}) \cdots \exp(-\omega_{u,par(u)}) \\
&= \frac{1}{z_i} \prod_{\forall (k,m) \in \mathbf{E}_{j,i}} \exp(-\omega_{k,m}).
\end{aligned} \tag{13}$$

Therefore, $P_{\mathcal{G}_T}(h_i = j) = \frac{1}{z_i} S_{\mathcal{G}_T}(\mathbf{E}_{j,i})$.

$\qquad\qquad\qquad\qquad\qquad\qquad\qquad\qquad\qquad\qquad\qquad\qquad\qquad\qquad\qquad\qquad\qquad\qquad\square$

**Lemma 2.** *Given a Markov Random Field on the tree $\mathcal{G}_T$, whose unary and pairwise terms are the Eq. 9 and the Eq. 10, respectively. When denoting node $i$ as the root of the tree $\mathcal{G}_T$ and node $v$ (distant) as one of the descendants node of node $u$ (nearby), the marginal probability of latent variable satisfies $P_{\mathcal{G}_T}(h_i = v) \le P_{\mathcal{G}_T}(h_i = u)$.*

*Proof.* Since the edge distance satisfies $\omega_{k,m} \ge 0$, the marginal probability of latent variable satisfies

$$
\begin{aligned}
P_{\mathcal{G}_T}(h_i = v) &= \frac{1}{z_i} S_{\mathcal{G}_T}(\mathbf{E}_{v,i}) \\
&= \frac{1}{z_i} S_{\mathcal{G}_T}(\mathbf{E}_{u,i}) S_{\mathcal{G}_T}(\mathbf{E}_{v,u}) \\
&= P_{\mathcal{G}_T}(h_i = u) \prod_{\forall (k,m) \in \mathbf{E}_{v,u}} \exp(-\omega_{k,m}) \\
&\le P_{\mathcal{G}_T}(h_i = u).
\end{aligned}
\tag{14}
$$

$\qquad\qquad\qquad\qquad\qquad\qquad\qquad\qquad\qquad\qquad\qquad\qquad\qquad\qquad\qquad\qquad\qquad\qquad\square$

**Lemma 3.** *Given a Markov Random Field on the tree $\mathcal{G}_T$, whose unary and pairwise terms are the Eq. 15 and the Eq. 10, respectively. When denoting node $i$ as the root of the tree $\mathcal{G}_T$ and node $v$ (distant) as one of the descendants of node $u$ (nearby), there exists a specific pair of $f(\cdot)$ and $\beta$, such that the marginal probability of latent variable satisfies $P_{\mathcal{G}_T}(h_i = v) > P_{\mathcal{G}_T}(h_i = u)$.*

$$
\phi_i(h_i, x_i) := \begin{cases} f(x_i) & h_i = i \\ \exp(-\beta) & h_i \ne i \end{cases}
\tag{15}
$$

*Proof.* For $h_i = v$, the marginal probability is

$$
\begin{aligned}
P_{\mathcal{G}_T}(h_i = v) &= \frac{1}{z_i} \exp(-\beta)^{|\mathbf{E}_{v,i}|} f(x_v) S_{\mathcal{G}_T}(\mathbf{E}_{v,i}) \\
&= \frac{1}{z_i} \exp(-\beta)^{|\mathbf{E}_{u,i}|} S_{\mathcal{G}_T}(\mathbf{E}_{u,i}) f(x_v) \exp(-\beta)^{|\mathbf{E}_{v,u}|} S_{\mathcal{G}_T}(\mathbf{E}_{v,u}) \\
&= P_{\mathcal{G}_T}(h_i = u) \frac{f(x_v)}{f(x_u)} \prod_{\forall (k,m) \in \mathbf{E}_{v,u}} \exp(-\omega_{k,m} - \beta).
\end{aligned}
\tag{16}
$$

When $\frac{f(x_v)}{f(x_u)} \prod_{\forall (k,m) \in \mathbf{E}_{v,u}} \exp(-\omega_{k,m} - \beta) > 1$, the marginal probability of latent variable satisfies $P_{\mathcal{G}_T}(h_i = v) > P_{\mathcal{G}_T}(h_i = u)$. $\qquad\square$