[Reviews · NeurIPS 2020]

Review 1

Summary and Contributions: This paper proposes a new mechanism to diffuse information inside a convnet across the spatial domain. The information diffusion is built on top of learned filters and spanning trees. This new layer is shown to improve results on semantic and instance segmentation tasks. The proposed method “LTF-V2” is presented as an evolution of the existing LTF-V1.

Strengths: - The proposed method builds on unusual ingredients (for typical 2020 computer vision paper) such as superpixels and spanning trees. I appreciate some papers trying different roads. - The results do show a small but consistent improvement. - The paper does include results comparing “gained quality versus model size/compute cost”, which is indeed a key aspect for this kind of contribution. In that sense the paper avoids the “adding complexity leads to better results” trap, by arguing that this is less computation complexity for a higher quality gain than competing methods.

Weaknesses: - The paper reads too much like LTF-V1++, and at some points assumes too much familiarity of the reader to LTF-V1. Since this method is not well known, I wish the paper was a bit more pedagogical/self-contained. - The method seems more involved that it needs to be. One would suspect that there is an underlying, simpler, principle that is propulsing the quality gains.

Correctness: Yes. I did not notice any particular logical flaw or strong experimental issue. Most results are reported on the dataset where the method was tuned (COCO val set) (“best-fit to test set” scenario), however the inclusion of Cityscapes test set results indicates that the method is able to generalize to other settings.

Clarity: The paper is reasonably well written. There are some minor issues with the English and a couple of equations could benefit from further clarification (e.g. Eq 8, see “additional feedback”); but nothing very bad. The main caveat is that the paper assumes too much familiarity with LTF-V1 and tree filtering in general (e.g. [23]). I suspect this might make the paper a harder read to our younger audience. I think adding a short paragraph highlighting the motivation behind tree filtering and its high-level pros and cons might help. Having said that, overall the paper is well written, and one can see a concerned effort on making the paper understandable.

Relation to Prior Work: Albeit there is no explicit “related work” section, I feel the paper reads fine as-is in this regard. The first paragraph gives a very brief overview of the ideas in the space of interest, and the experimental results section does include a comparison with a reasonable set of alternatives. I would have appreciated including a more direct comparison to using dilated convolutions [10], which I understand as having become the “standard” approach to handle large context. It is not included explicitly in tables 1 nor 6 (forcing the reader to compare tables across papers). Ideally this should be fixed in the camera ready version.

Reproducibility: Yes

Additional Feedback: Here are some detailed comments and questions: L5: I do not believe that “we give the quantitative analysis by reformulating” is proper nor correct English. Please rephrase. L28: to be the -> to be its L63: mention what you have in mind as “guidance”. For readers unfamiliar with tree filtering this comes out of the blue. L85: computational efficiency is good in CPUs, what about GPUs or “tensor processing units” ? L91: I suspect the word “quantitatively” is not being used correctly. Please rephrase. L131: Why force a closed-form solution ? Would it not be better to just use a grid structure and obtain an approximate solution ? (Would be extra-nice to have experiments on this regard). Eq 8: would be good to link back to Eq 6. As is, the relation seems opaque. L158: Linear on what ? number of pixels ? L161: determinate -> deterministic L169: so the “the grouping strategy” is similar to what is used in ResNeXt ? Please add a relevant citation. Sec 3.1: Consider renaming the section as “Instance segmentation experiments” (similar for 3.2 title). Sec 3.1.1: to better link with table 1 (and paragraph at L218), I suggest mentioning Resnet 50 here. L210: Which table does this refer to ? Which benchmark ? L246: The “superiority” claim is somewhat loose, since many methods obtain better results. Maybe specify more the constraints ? (fine-annotations only, only imagenet pre-training, Resnet-101 backbone only ?)


Review 2

Summary and Contributions: Learnable Tree Filter (LTF) is an efficient yet effective feature transformation module that allows neural networks to enlarge receptive field while considering semantic structure of image. This paper revises LTF by interpreting it as a Markov Random Field (MRF), introducing learnable unary potentials of the MRF, and learning networks with spanning trees sampled in a stochastic manner. These components allow LTF to enlarge its receptive field more effectively and contribute to the performance improvement in a several object detection and segmentation tasks.

Strengths: (1) This paper presents a well principled method that addresses an important problem shared by many different visual recognition architectures. It is technically sound and has a great potential. (2) The proposed LTF module allows a standard semantic segmentation network to achieve state of the art on the Cityscape dataset. (3) The proposed method has been evaluated extensively on two different recognition tasks. In addition, results of the thorough ablation studies suggest that every component of the new LRF module contributes to the performance improvement, and that the module can be applied to a variety of recognition architectures. (4) The paper is not very novel, but I believe its contributions are sufficient to be introduced in NeurIPS. As the paper is about an extension of existing technique, its inherent novelty is limited. However, the analysis is technically sound and newly introduced components are all reasonable, sufficiently different from the original ones, and empirically effective (sometimes the improvement is marginal though).

Weaknesses: Performance improvement by the proposed method is sometimes marginal. Although the new version of LTF, proposed in this paper, achieved state of the art on the challenging Cityscape dataset, the improvement by replacing the previous version of LTF with the new one seems not that great on the MS-COCO dataset. Also the improvement becomes smaller when stronger backbone networks are adopted.

Correctness: Could not find any critical flaws in technical aspects.

Clarity: The paper is written clearly, thus easy to understand and reproduce the proposed module.

Relation to Prior Work: There is no related work section and only very closely related papers have been introduced shortly in the manuscript. I know the manuscript is already quite dense, but would like to recommend to present related work in more detail, especially including self attention mechanisms and non-local neural network.

Reproducibility: Yes

Additional Feedback: The rebuttal well addresses my concerns. Hope the authors reflect the valuable comments by the reviewers (especially, comparisons to previous work on exploiting context or enlarging receptive field effectively in semantic segmentation, as commented by R8).


Review 3

Summary and Contributions: This work proposes the Learnable Tree Filter V2 (LTF-V2) that improves LTF-V1 in multiple ways. First, it proposes a new learnable unary term by reformulating the geometric relations between features as an MRF. Second, it replaces the non-differentiable spanning tree algorithm with a learnable one. The proposed LTF-V2 is evaluated on two benchmark datasets, MSCOCO and CityScapes, and show better performance than the LTF-V1.

Strengths: + The proposed LTF-V2 can be useful to embed context information for feature learning with linear complexity. + The LTF-V2 clearly improve its previous version (LTF-V2) in the multiple tasks that require context modeling, including object detection, instance segmentation and semantic segmentation. + The performance of LTF-V2 is competitive on both COCO 2017 val set and Cityscapes val set.

Weaknesses: 1. The technical contribution of this work is highly limited. (1) This work solely focuses on improving one particular algorithm named LTF-V1. - It first identifies some specific issues of this model and then proposes the ideas to resolve them. I can clearly understand that LTF-V2 is a better model than its original version, but cannot find any generality to a broader set of algorithms in this area of research. - Simply speaking, the two key technical contributions of this work, a new unary term and a new spanning tree algorithm, is meaningful only with LTF-V1. (2) Moreover, the two contributions seem rather incremental. The first one simply replaces Eq.(2) with Eq.(8) using the formulation of MRF, and the second is a simple randomization of trees as described in section 2.4. 2. Experimental comparison should be updated to include more recent methods. (1) In Table 1 and 6, some recent works have been omitted for comparison. - For example, for the COCO 2017 experiments, the following papers may need to be cited and compared. a. D. Ruan et al. Linear Context Transform Block. AAAI 2020. b. J. Yan et al. FAS-Net: Construct Effective Features Adaptively for Multi-Scale Object Detection. AAAI 2020. - For the Cityscapes experiments, consider a. Q. Hou et al. Strip Pooling: Rethinking Spatial Pooling for Scene Parsing, CVPR 2020. b. C. Yu et al. Context Prior for Scene Segmentation, CVPR 2020. (2) The performance gaps with other state-of-the-art models are nearly marginal as shown in Table 1 and 6. The statistical significance of the results should be discussed. 3. Other comments (1) The results of Table 1 are not discussed at all in the text, even though it is one of key results of this work. (2) Fig. 1 can be improved to clarify the enhancement of LTF-V2 over LTF-V1. (3) Alg. 1 contains a simple and basic randomized algorithm for sampling a spanning tree. It could be removed from the main draft.

Correctness: The key claims and the proposed method of this work seems correct.

Clarity: This paper is written well enough.

Relation to Prior Work: Yes.

Reproducibility: Yes

Additional Feedback: <Post-rebuttal review> My initial main concerns were two-fold: (i) unconvincing novelty and (ii) limited experiments. The authors’ response is sufficiently persuasive for (i) but not much for (ii). The rebuttal simply promises the update of the final draft without no specific ground (e.g. For Q2 and Q3). I’d like to raise my score from 4 to 5.


Review 4

Summary and Contributions: This work improves over learnable tree filter version 1, which balances between learning long-range dependencies and object details preserving. The propsoed learnable tree filter version 2 relaxes the geometric constraint and resovles the non-differentiable spanning tree construction of the version 1. When tested on COCO and Cityscapes with Res-50 FPN backbone, it achieves the state-of-the-art AP and mIoU scores, respectively.

Strengths: The proposed work improves upon the LTF-V1 by reformulating it as a MRF, which relaxes the geometric constraint of V1. The proposed learnable spanning tree is differentiable and can be inserted in any stages in typical networks (e.g., resnets) and be trained end-to-end. I achieves state-of-the-art results on COCO and Cityscapes benchmarks. The module is efficient compared to the prior arts, adding relatively less number of parameters.

Weaknesses: The visualization in Figure 3 is hard to interpret and the explanation is not intuitive enough. I am curious if the proposed methods learns position sensitivity of the query pixels. For example, in Figure 3, how would multiple query pixels in the same image result in different pairwise dependencies ? I would suggest the authors to perform panoptic segmentation, which can show both instance-aware task and semantic segmentation task on a single dataset: COCO or Cityscapes, to further emphasize the effectiveness of the algorithm. L203: qualitative -> quantitative

Correctness: The claims and empirical methodology is correct.

Clarity: The paper is well written.

Relation to Prior Work: It is clearly discussed.

Reproducibility: Yes

Additional Feedback: ----- I read the other reviews and rebuttal. I am leaning towards my original rating.

[Author Response · NeurIPS 2020]

# Author Rebuttal for NeurIPS 2020 Submission #2238

We thank all the reviewers for their valuable comments and suggestions. To improve readability, we promise to add a section of the background and carefully revise the manuscript before the final submission. We respond to the main concerns as follows. All the source code will be released to the community soon.

## Response to Reviewer #4

**Q1:** *Some suggestions for improving the writing quality.*

**A1:** Thanks for your valuable suggestions! We will revise the manuscript carefully according to your comments.

**Q2:** *What about the computational efficiency on GPUs or "tensor processing units"?*

**A2:** The empirical runtime on GPUs is very low, which has been reported in Fig.4 of the supplementary material. We optimize the CUDA kernel by parallelizing the algorithm along with batches, channels, and nodes of the same depth.

**Q3:** *Why force a closed-form solution? Would it not be better to just use an approximate solution on grid structures?*

**A3:** Compared with the iterative optimization process for grid structures, the closed-form solution ensures that the LTF-V2 has a deterministic and negligible computational complexity to obtain the global receptive field. Besides, the low-level applications of tree filter (*e.g.*, stereo matching and image denoising) demonstrate the competitive performance against many energy-optimization based methods. We will add some related experiments in the final version.

## Response to Reviewer #7

**Q1:** *Performance improvement on the COCO dataset is smaller than that on the Cityscapes dataset.*

**A1:** This is a valuable question. It seems to be a common phenomenon of related methods, *e.g.*, Non-Local [11] and CCNet [12]. We think the reason may come from different metrics for different tasks and the more complex scenarios on the COCO dataset. Nevertheless, compared with competing methods, the LTF-V2 achieves more quality gains. Besides, as shown in Tab. 8, our method achieves 2.9% mIoU absolute gains over baseline without bells-and-whistles.

## Response to Reviewer #8

**Q1:** *Some concerns about the technical contribution of this work.*

**A1:** The traditional tree filter [23] already has great impacts on many low-level computer vision tasks, owing to its structure-preserving property and high efficiency. This paper further releases its representation potential by relaxing the geometric constraint. Albeit the modification of the proposed module seems small in form, it has significantly different properties from the original module (refer to Fig.3) and enables fully end-to-end training, which could obtain consistent improvements with negligible computational and parametric overheads. These properties have great potentials which allow our method and principle to extend to other complex tasks with large-number nodes, *e.g.*, replacing the attention module of transformer for natural language processing and enhancing sequential representation for video analysis. We will add more details and revise the "broader impact" section in the final version.

**Q2:** *Experimental comparison should be updated to include more recent methods.*

**A2:** Thanks for your suggestion. In fact, according to the comparative analysis, we find that our approach is superior to the listed methods when using the same backbone. We will add more recent methods in the final version.

**Q3:** *Some concerns about the performance gaps with other state-of-the-art models.*

**A3:** Our experiments are constructed not only to achieve top performance, but also to demonstrate that a higher quality gain can be achieved with less complexity. Therefore, our method does not adopt additional enhancements (*e.g.*, GCNet [15] is applied in each bottleneck and DANet [46] uses the "multi-grid" operation). We will give more competitive results with these enhancements in the final version.

## Response to Reviewer #9

**Q1:** *I am curious if the proposed methods learns position sensitivity of the query pixels.*

**A1:** Yes. As shown in Eq.2, for each query pixel $i$ and key pixel $j$, the pairwise affinity $S_{\mathcal{G}_T}(E_{i,j})$ is regenerated by accumulating all the edge weights along the path from $i$ to $j$. Since different query pixels have different paths to the same key pixel, the corresponding affinities can be inconsistent. Therefore, the pairwise term (refer to Eq.6) has the position-sensitive property. We will clarify it in the final version.

Table 8: The ablation study for panoptic segmentation on COCO 2017 *val* set.

| Model | Backbone | Schedule | LTF-V2 | PQ | SQ | RQ | mIoU | mACC | $AP_{det}$ | $AP_{seg}$ |
|---|---|---|---|---|---|---|---|---|---|---|
| Panoptic FPN | ResNet-50 | 1x | ✗ | 39.6 | 77.8 | 48.6 | 41.6 | 52.3 | 37.6 | 34.7 |
| | | | ✓ | **42.0** | **79.0** | **51.1** | **44.5** | **56.5** | **39.5** | **36.1** |

**Q2:** *I would suggest the authors to perform panoptic segmentation.*

**A2:** Thanks for your suggestion. We construct an experiment based on Panoptic FPN [CVPR 2019] on the COCO dataset. The LTF-V2 is adopted after "Stage3", "Stage4" and "Stage5" of the backbone. As shown in Tab. 8, the result further demonstrates the effectiveness and generalization of our method. We will add more results in the final version.

[Meta-Review · NeurIPS 2020]

The paper presents an extension of learnable tree filter method (LTF). On the positive side, the proposed method looks reasonable and it leads to improved performance over the baselines. However, on the weak side, the proposed contributions are fairly incremental, and performance improvement is somewhat marginal (over LTF-v1 and other SOTA methods), and comparisons against some state-of-the-art methods are missing. Overall, given the strong support from three reviewers out of four, I would give a chance for the authors to address the issues. If accepted, the authors should revise the paper according to their rebuttal.